**Key words:**
Microdroplet chemistry; DNA; Reaction Rate Acceleration; Deoxyribonuclease; SARS-CoV-2Orf1ab

**Author for correspondence:**
Richard N. Zare,
E-mail: rnz@stanford.edu

# Ultrafast enzymatic digestion of deoxyribonucleic acid in aqueous microdroplets for sequence discrimination and identification

Xiaoqin Zhong[1,2] , Hao Chen[3] and Richard N. Zare[1,4]

[1]Department of Chemistry, Fudan University, Shanghai, China; [2]Department of Environment and Chemical Engineering, Shanghai University, Shanghai, China; [3]Department of Chemistry & Environmental Science, New Jersey Institute of Technology, Newark, NJ, USA and [4]Department of Chemistry, Stanford University, Stanford, CA, USA

## Abstract

We report the use of aqueous microdroplets to accelerate deoxyribonucleic acid (DNA) fragmentation by deoxyribonuclease I (DNase I), and we present a simple, ultrafast approach named DNA fragment mass fingerprinting to discriminate different DNA sequences by comparing their fragment mass patterns. DNA fragmentation in tiny microdroplets, which was produced by electrosonically spraying (+3 kV) a room temperature aqueous solution containing 10 µM DNA and 10 µg ml$^{-1}$ DNase I from a homemade setup, takes less than 1 ms. High differentiation/identification fidelity could be obtained by applying a cosine correlation measure for similarity assessment between two fragment mass patterns, which compares both mass-to-charge ratios ($m/z$) with an error tolerance of 5 ppm and the peaks' relative intensities. A single-nucleotide mutation in the sequence of bases, as exemplified by the sickle cell anemia mutation, is differentiated by setting a cutoff value of similarity at 90%. The order change of two adjacent bases in the sequence could still be well discriminated with a similarity of only 62% between the fragment mass patterns of the two similar sequences, which have the same molecular weights and thus cannot be differentiated by gel electrophoresis or direct mass detection by mass spectrometry. Compared to traditional genotyping methods, such as quantitative real-time polymerase chain reaction, the identification process with our approach could be completed within several minutes without any other expensive and complicated reagents or experimental steps. The potential of our approach for convenient and fast microbe genetic discrimination or identification is further demonstrated by differentiating the Orf1ab gene fragments of two similar coronaviruses with a very high sequence homologous rate of 96%, SARS-CoV-2 and bat-SL-CoVZC45, with a similarity of 0% between their fragment mass patterns.

## Introduction

In the past few years, it has been demonstrated that reactions run in microdroplets can be remarkably accelerated compared to the same reactions in bulk solvents (Wei *et al.*, 2020). Almost all these reactions have involved small organic molecules, but recently this marked acceleration has been demonstrated for several protein systems, such as the unfolding of cytochrome *c* (Lee *et al.*, 2015), and the enzymatic digestion in aqueous microdroplets of proteins and therapeutic antibodies (Zhong *et al.*, 2020; Zhao *et al.*, 2021). These results have motivated us to examine the enzymatic digestion of nucleic acids in aqueous microdroplets. Here, we report several practical examples that demonstrate how the digestion of deoxyribonucleotide acids (DNAs) can be accelerated in water microdroplets to provide simple, fast sequence differentiation and identification.

Genotyping is the process of determining an individual's DNA sequence using biological assays and how this sequence compares to another individual's sequence or a reference sequence (Gurbel *et al.*, 2010). This task is important in research on genes and gene variants associated with disease. Restriction fragment length polymorphism (RFLP) is considered to be the simplest and earliest genotyping method, which performs digestion on single or multiple genomic regions with different restriction endonucleases and determines fragment lengths through a gel assay to ascertain whether the enzymes cut the expected restriction sites (Saiki *et al.*, 1985; Pejic *et al.*, 1998; Dai and Long, 2015). Unfortunately, the requirement for specific endonucleases, and the slow nature of gel assays make RFLP a poor choice for high throughput analysis. In addition, the low resolution offered by gel assays often fails to resolve two fragments with very similar molecular weights. Polymerase chain reaction (PCR) is now widely used to specifically amplify and determine a chosen genomic region by designing a PCR primer and a probe labelled with a flaring group to report the existence of the chosen genomic region or not, which complicates the genotyping process and is less cost-efficient (Van Elden *et al.*, 2001; Gibson, 2006; Baker, 2010). In contrast to these traditional methods mentioned above, most of which take several hours and require expensive and complicated reagents, we present an alternative approach that makes only

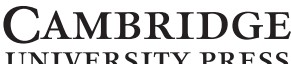

use of a DNA exonuclease to digest DNA samples in aqueous microdroplets within microseconds. When the microdroplets are analysed with a high-resolution mass spectrometer, this procedure normally takes several minutes and differentiates two fragments with a mass shift larger than 5 ppm.

Since its existence, peptide mass fingerprinting (PMF) has proven to be a fast, sensitive approach for the identification of an unknown protein, which usually relies on enzymatic digestion of the protein followed by accurate mass determination of the resulting peptide fragments by mass spectrometry (MS). The masses of the peptides constitute a fingerprint of the protein under investigation, which is compared with the fingerprint that generated from an *in silico* digest of all entries in a protein database to obtain the best matching candidate protein (Henzel *et al.*, 1993; Pappin *et al.*, 1993). Similarly, an unknown RNA or DNA should also be amenable to the same identification approach as PMF of proteins. Practical RNA characterisation by nucleotide-specific cleavage followed by MS analysis has been first performed by Crain and McCloskey to search for post-transcriptional modifications in known sequences (Kowalak *et al.*, 1993; Kirpekar *et al.*, 2000). Various typings were further reported based on a common strategy: a copy of the genomic region of interest is obtained by PCR followed by *in vitro* transcription into RNA, and the enzymatic digestion of the RNA will generate an origin-specific digestion pattern that can be visualised by MS (Limbach *et al.*, 1995; Von Wintzingerode *et al.*, 2002; Hartmer *et al.*, 2003). However, DNA characterisation with this similar approach was seldom reported, which holds several potential advantages over RNA characterisation, including no need of *in vitro* transcription, much less cost of syntheses, higher sensitivity of MS detection, and better chemical stability of DNA against self-degradation for longer storage. Based on previous protein-involved work, in the present study, we continue to expand the application of microdroplets to nucleic acid analysis to develop what we call DNA fragment mass fingerprinting (DMFP).

In water microdroplets produced by a homemade electrosonic spray ionisation (ESSI) setup, ultrafast digestion of 10 μM DNA with 10 μg ml$^{-1}$ deoxyribonuclease I (DNase I) could be completed within microseconds compared to traditional bulk reaction that normally takes 1–2 h. Combined with on-line detection by a high-resolution orbitrap mass spectrometer, accurate sequence discrimination and identification could be completed within several minutes by automatically assessing the similarity between the fragment mass patterns of two DNA samples with a software based on the cosine correlation measure. This procedure compares both mass-to-charge ratios (*m/z*) with an error tolerance of 5 ppm and also the peaks' relative intensities to increase the identification fidelity. Potential practical application in discriminating two similar sequences was demonstrated by a sickle cell anemia diagnosis caused by only a single-nucleotide mutation. Two designed sequences with the same molecular weights but an order change of two adjacent bases in the sequence, which could not be resolved by gel electrophoresis or direct mass detection by MS, were successfully discriminated with a similarity of only 62% between their fragment mass patterns. The order interchange was chosen just to illustrate the power of this method for genotyping. Using our approach, we compared the Orf1ab gene fragments of two similar coronaviruses with a very high homologous rate of 96%, SARS-CoV-2 and bat-SL-CoVZC45. We found a similarity of 0% between their fragment mass patterns, showing the superiority of microdroplets coupled with high-resolution MS detection to achieve simple, fast and economical microorganism genetic discrimination

or identification. Overall, the results suggest to us that DMFP may have wide applicability.

## Materials and methods

### Microdroplet-MS with ESSI

A stream of microdroplets was generated by infusing an aqueous sample solution containing DNA molecules (10 μM) and DNase I (10 μg ml$^{-1}$; 0.3 μM) in 5 mM ammonia acetate (NH$_4$OAc, pH 8) containing 0.1 mM MgCl$_2$ and 0.1 mM CaCl$_2$ with a syringe at a flow rate of 10 μl min$^{-1}$ into a homemade sprayer. The enzyme of DNase I (from bovine pancreas, molecular biology grade, activity ≥500 kU/ mg) was obtained from Sangon (Shanghai, China). All the DNA samples were synthesised by Sangon. Ammonia acetate (analytical reagent, ≥98%), calcium chloride (analytical reagent, ≥98%) and magnesium chloride (analytical reagent, ≥98%) were obtained from Titan (Shanghai, China). Deionised water (18.2 MΩ cm) was prepared by the Milli Q purification system (Millipore Advantage A10; Sigma-Aldrich, St. Louis, MO) and used in all aqueous solutions.

The sample solution was sprayed from the tip of a fused silica capillary (148 μm o.d., 50 μm i.d., Polymicro Technologies, Shanghai, China) of the homemade sprayer and assisted by a nebulising gas of dry N2 with a pressure of 120 psi. By placing the sprayer in front of a high-resolution mass spectrometer (LTQ Orbitrap Elite, Thermo Scientific, San Jose, CA) at a proper position, the microdroplets were directed into MS for real-time analysis when applying a positive high voltage (+3 kV, BOHER HV, Genvolt, Bridgnorth, UK) to the sprayer. The MS inlet capillary was always maintained at 275 °C and capillary voltage at 0 V. No other source gases were used when digestion was performed in microdroplets.

For control tests, DNA was also digested using a standard procedure. 10 μM DNA was incubated with 10 μg ml$^{-1}$ of DNase I in a 5 mM NH$_4$OAc buffer, pH 8, under room temperature (RT) for 1 h and further submitted to standard electrospray ionisation (ESI) MS analysis.

### Commercial ESI-MS

For the analysis with standard ESI-MS, the samples were also directly infused with a syringe at the flow rate of 10 μl min$^{-1}$ and sprayed from a commercial heated ESI probe with a needle of around 500 μm in inner diameter fitted for a high-resolution mass spectrometer (LTQ Orbitrap Elite, Thermo Scientific, San Jose, CA). The spray was assisted with a sheath gas flow of 10 arbitrary units (10 psi). The temperature of the MS inlet capillary was set at 275 °C and the ESI voltage was set as +3 kV.

### DNA sequence discrimination

After intensity normalisation of a mass spectrum, only multiple-charged peaks ($z \geq 2$) with a signal-to-noise ratio (S/N) of at least 3 and relative intensity of at least 5% were extracted and reconstructed into a fragment mass pattern. Data analysis and conversion into exact mass list were performed by Xcalibur Qual Browser (ThermoFisher Scientific). The fragment mass patterns were plotted by Origin Pro (Version 8.5 for Windows). The matching level between the fragment mass patterns from two samples was evaluated by calculating the pattern similarity score using the cosine correlation algorithm automatically with an open-source software (http://bacteriams.com; Yang *et al.*, 2017). A tolerance of 5 ppm was chosen. A score ≥90% was the threshold for successful identification.

## Results and Discussion

### *Performance optimisation of microdroplet-MS*

Fig. 1 presents a schematic drawing of the experimental apparatus for microdroplet-MS, which is similar to that used in our previous work (Zhong *et al.*, 2020).

Briefly, microdroplets containing 10-μM DNA sample and 10-μg ml$^{-1}$ enzyme of deoxyribonuclease I (DNase I) dissolved in 5-mM buffer of ammonium acetate were generated by a home-made electrosonic sprayer in which a sheath of rapidly flowing dry N$_2$ gas at 120 psi surrounded an inner capillary held at typically +3 kV. The initial droplet size generated by ESSI was previously measured by a laser particle analyser to be around 9 μm (Zhong *et al.*, 2020).

A commercially synthesised DNA sequence, 5′-TAGTCAGCTACGGCTAGA-3′, is used as a simple model for system optimisation, especially the travel distance of microdroplets. How the travel distance influences the sample digestion extent has been explained in previous work (Zhong *et al.*, 2020). Briefly, during the flight into the mass spectrometer, the sample digestion in microdroplets is accelerated significantly with possible mechanisms as described in the final discussion part. We suppose that the initial droplet size is critical for reaction acceleration rather than the aqueous droplet shrinkage from evaporation. Gomez and Tang (1994) experimentally find that it takes approximately 547 μs for a heptane droplet with a diameter of 4.7 μm to evaporate to the

point of first droplet Coulomb explosion. For aqueous droplets with an initial size of 9 μm produced by ESSI in our case, the time required for the first droplet fission to occur is of course longer than 547 μs owing to the lower vapour pressure of water compared to heptane. This time is also longer than the travel time of microdroplets before entering the MS inlet due to the very high flight velocity of 80 m s$^{-1}$ with our setup (Lee *et al.*, 2015). Once inside the heated inlet, the microdroplet evaporates quickly caused by the heated MS inlet (275 °C) and the reaction stops (Zhong *et al.*, 2020). Therefore, a longer travel distance of microdroplets result in more time for digestion. We determined the extent of digestion by varying the travel distance between the sprayer tip and the MS inlet. For easy comparison, part of the model DNA's whole fragment mass spectrum from *m/z* 900–1500 with two fragments' peaks close to the original DNA peak was extracted, as shown in Fig. 2. We found that the digestion extent of DNA in microdroplets varied significantly by changing the travel distances of microdroplets from 2 to 25 mm with increasing relative intensities of the fragments and decreasing relative intensity of the original DNA molecule, as marked by dashed square lines in Fig. 2*a–d*. Most of DNA was digested with only a tiny peak from the original DNA molecule seen in Fig. 2*c* at a travelling distance of 25 mm. Although higher extent of digestion could be achieved by further increasing the travel distance, the entire MS signal decreases, and therefore the distance of 25 mm from the sprayer tip to MS inlet was adopted for all the following tests. The digestion time in the microdroplets was

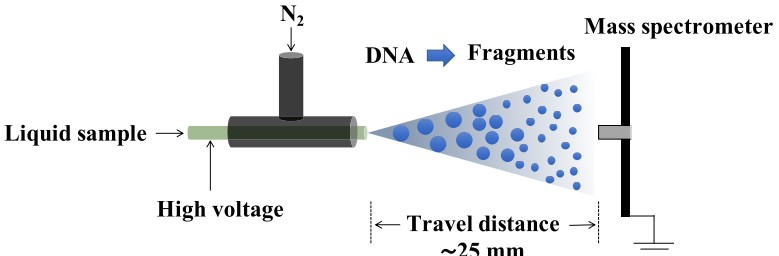

**Fig. 1.** Schematic of the experimental apparatus for DNA fragmentation by microdroplets coupled with mass spectrometry (MS). The inner capillary has an i.d. of 50 μm and an o.d. of 148 μm to which a high voltage (+3 kV) is applied.

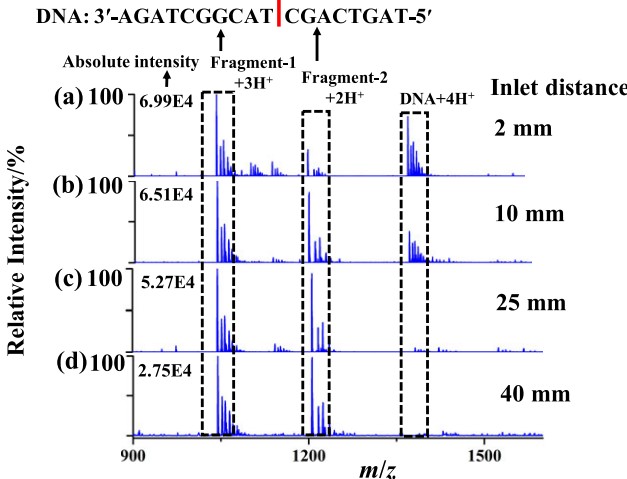

**Fig. 2.** Mass spectra of 10-μM DNA (5′-TAGTCAGCTACGGCTAGA-3′) in 5-mM aqueous ammonium acetate sprayed by the homemade setup: (*a–d*) digested with 10-μg ml$^{-1}$ DNase I (corresponding to 0.3 μM) at different travel distances between the sprayer tip and MS inlet for 2, 10, 25 and 40 mm, respectively. The relative peak intensities of two fragments with mass-to-charge ratios (*m/z*) at 1044.8 and 1205.2 were compared to that of the quadruple-charged DNA molecule with *m/z* at 1381.7, as marked by dashed square lines. The short red line shows the cleavage site in the model DNA sequence.

determined to be approximately 0.3 ms based on the previously reported microdroplet velocity of 80 m s$^{-1}$ under these conditions (Lee *et al.*, 2015).

### Base-preferential cleavage by DNase I

DNase I is a nuclease that cleaves single-stranded or double-stranded DNA with no absolute base or sequence specificity, yielding 5′-phosphate-terminated polynucleotides with a free hydroxyl group on 3′-position (Samejima and Earnshaw, 2005). In this work, we studied whether the DNase I from bovine pancreas (Sangon) holds some levels of cleavage preference to the four kinds of DNA bases.

We designed a series of different DNA sequences, as listed in Supplementary Table S1. From the results summarised in Supplementary Table S1, the same experimental conditions result in very different fragmentation patterns for different DNA sequences. The DNase I shows highest preference for cleaving at the thymine (T) base. We found that a sequence containing pure T was totally fragmented with no original DNA peaks, as shown in Supplementary Fig. S1. In sharp contrast, the sequences containing pure A or pure C could not be fragmented, even after being incubated with DNase I overnight, as shown in Supplementary Figs. S2 and S3. We also inserted three other kinds of bases, G, T and C, into the sequence of pure A and fragments were still not found in the mass spectra. The same conclusion also applies for the sequence of pure C. These results demonstrate that the bases of A and C are strongly resistant to fragmentation by DNase I. We can expect that the existence of a large amount of A or C in a DNA sequence, will protect this DNA sequence from being digested by DNase I. Although absolute base or sequence specificity of DNase I seems not to have been reported previously, in the current study, we are still puzzling over the possible reasons that cause the strong preference of DNase I to T.

### Stability and reproducibility of DNA fragment mass pattern

Our work aims to apply microdroplets for the simple and fast DMFP, of which the theoretical basis assumes that the enzymatic digestion of a DNA sample with a specific sequence will produce a distinct fragment mass pattern that should be very stable, reproducible and differentiable. Fig. 3 shows the results of the two tests with the model DNA sample (5′-TAGTCAGCTACGGCTAGA-3′) under fixed experimental conditions, which allows us to evaluate the stability and reproducibility. The testing was performed, including five parallel samples from the same batch, and three samples from different batches on different days. The representative mass spectra of the two tests appeared similarly as shown in Fig. 3a,b.

To assess the similarities of the mass spectra in Fig. 3a,b more accurately, a general comparison process was performed as shown in Fig. 4a. To ensure the peaks coming from DNA fragments, avoid the interference of background signals and simplify the comparison process, multi-charged DNA fragments with a relative intensity larger than 5% were first extracted from the original mass spectrum to reconstruct into a fragment mass pattern. An often-used cosine correlation algorithm defined as the equation in Fig. 4b, where $y$ is the normalised intensity of a peak appearing in both spectrum $i$ and spectrum $j$ (an identical peak), $l$ is the number of identical peaks in the two spectra, $Y$ is the normalised intensity of a peak appearing in a spectrum and $n$ is the number of peaks in a spectrum, was applied for the similarity scoring between two fragment mass patterns by considering mainly differences in relative intensities between the common peaks from two fragment mass patterns and the number of common peaks. Peaks appearing in different mass patterns with $\triangle(m/z)/(m/z) <5$ ppm were considered as identical peaks according to the highest resolving power of orbitrap analysis. An open source software (http://bacteriams.com), which was originally designed for bacteria identification and fingerprinting, is used for the similarity assessment of different DNA fragment mass patterns (Yang *et al.*, 2017).

The similarity score varied from 97 to 99% and 95 to 99% for the five repetition samples from the same batch in Fig. 3a and three samples from different batches on different days in Fig. 3, respectively. These results demonstrate the high stability and reproducibility of the fragment mass pattern of a specific DNA sequence under a fixed experimental condition and confirm the feasibility of our approach for DNA fragment mass fingerprinting. The

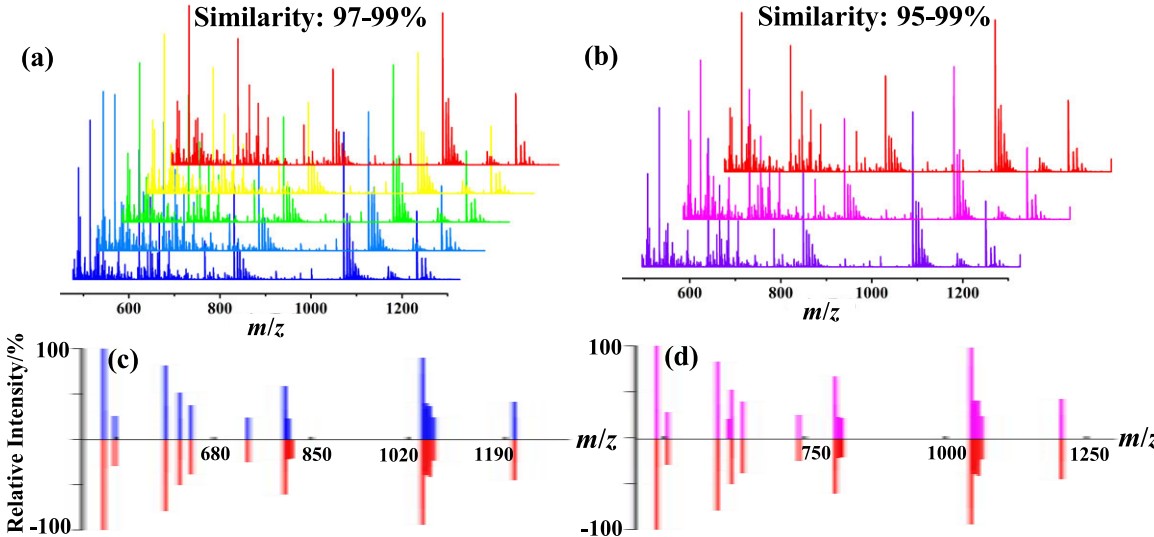

**Fig. 3.** Stability and reproducibility of fragment mass pattern of 10-μM DNA having the sequence of 5′-TAGTCAGCTACGGCTAGA-3′ in 5-mM aqueous ammonium acetate containing 10-μg ml$^{-1}$ DNase I sprayed by the homemade setup. Representative examples of mass spectra of: (*a*) five parallel samples from the same batch and (*b*) three samples from different batches on different days. Representative examples of fragment mass pattern comparison between: (*c*) the parallel samples from the same batch and (*d*) the samples from different batches on different days.

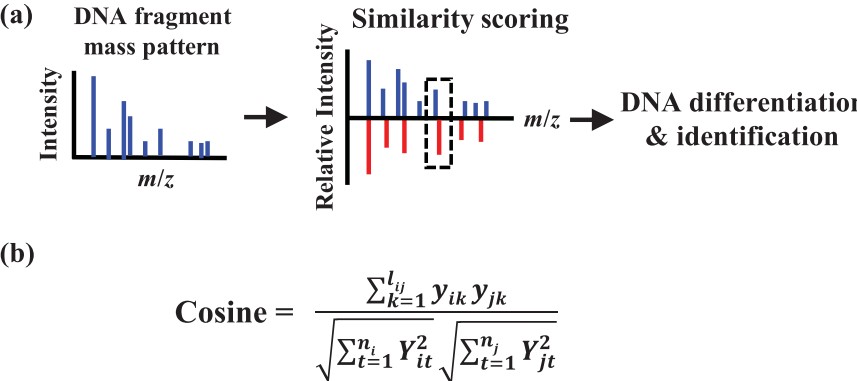

**(a)** DNA fragment mass pattern → Similarity scoring → DNA differentiation & identification

**(b)**

$$\text{Cosine} = \frac{\sum_{k=1}^{l_{ij}} y_{ik} y_{jk}}{\sqrt{\sum_{t=1}^{n_i} Y_{it}^2}\sqrt{\sum_{t=1}^{n_j} Y_{jt}^2}}$$

**y:** the normalized intensity of a peak appearing in both spectrum i and j
**l:** the number of identical peaks in the two spectra
**Y:** the normalized intensity of a peak appearing in a spectrum
**n:** the number of peaks in a spectrum

**Fig. 4.** (*a*) The general process of the DNA differentiation/identification based on the similarity assessment between two fragment mass patterns and (*b*) equation of cosine correlation measure used for similarity assessment.

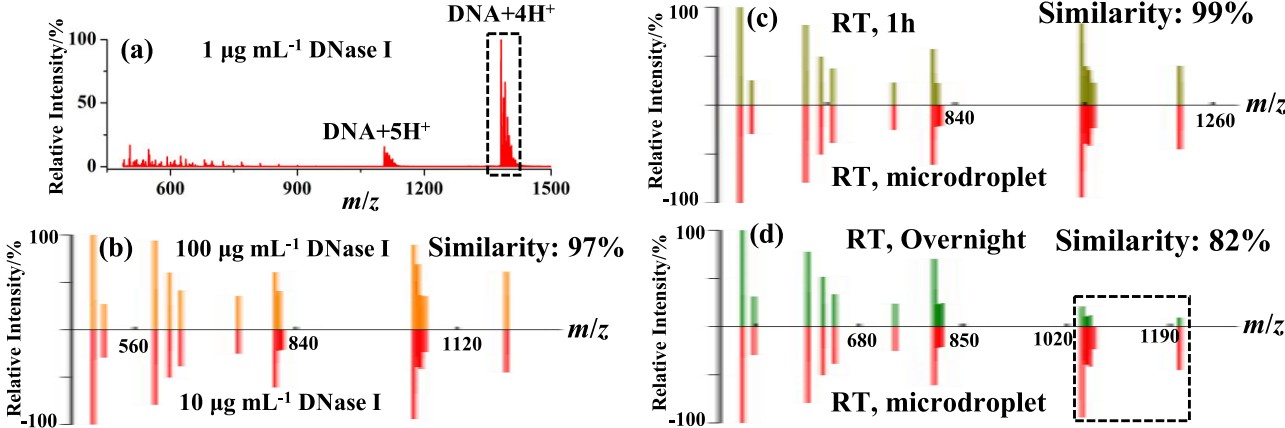

**Fig. 5.** 10-μM DNA (5′-TAGTCAGCTACGGCTAGA-3′) in 5-mM aqueous NH4OAc sprayed by the homemade setup under different experimental conditions: (*a*) mass spectra digested with 1-μg ml$^{-1}$ DNase I. Fragment mass pattern comparison between: (*b*) the two samples digested with 10-μg ml$^{-1}$ (red) and 100-μg ml$^{-1}$ (orange) DNase I, (*c*) the two samples digested with 10-μg ml$^{-1}$ DNase I for 1 h in bulk phase at room temperature (RT; olive green) and in microdroplets (red), and (*d*) the two samples digested with 10-μg ml$^{-1}$ DNase I overnight at RT (green) and in microdroplets (red).

similarity comparisons between two fragment mass patterns of the two tests in Fig. 3*a,b* are representatively shown in Fig. 3*c,d*, respectively.

Different experimental conditions, including the DNase I concentration and digestion time, were examined to see whether they will have significant impacts on the fragment mass pattern of the model DNA sample. Fig. 5b suggests to us that increasing the DNase I concentration to 100 μg ml$^{-1}$ has no or little change on the fragment mass pattern of the model DNA sample with a high similarity of 97% compared to the fragment mass pattern obtained using 10 μg ml$^{-1}$ DNase I.

But using 1 μg ml$^{-1}$ DNase I resulted in almost no fragments, as shown in Fig. 5*a* which means most DNA molecules failed to be digested by such a low concentration of DNase I and 10 μg ml$^{-1}$

DNase I is thus employed for the following applications. Compared to the fragment mass pattern obtained using microdroplets for digestion of the model DNA sample, the bulk digestion of the same DNA solution at RT for 1 h followed by detection with ESI-MS presents a very similar fragment mass pattern with a similarity score of 99% in Fig. 5*c*. However, after overnight digestion at RT,

the similarity score decreases to 82% with much lower relative intensities of big fragments as marked by the dashed square line in Fig. 5*d*.

With almost all similarity scores >90% among the fragment mass patterns of the same DNA sample from different repetitions and at different DNase I concentration, we can therefore assume that a DNA sample under investigation can be identified as the reference one when the similarity score between their fragment mass patterns is ≥90% (Zhu *et al.*, 2016). In our study, a score of ≥90% was considered as the threshold for a successful identification of DNA sequence.

### Discrimination of human hemoglobin beta chain (HBB) gene segment and its variants

Using microdroplets as an ultrafast digestion tool, our purpose in this study is to develop an approach, we call DMFP, for DNA sequence discrimination and identification. The findings discussed above present good stability, reproducibility of the DNA fragment mass pattern, and lay the foundation for DMFP. To examine the

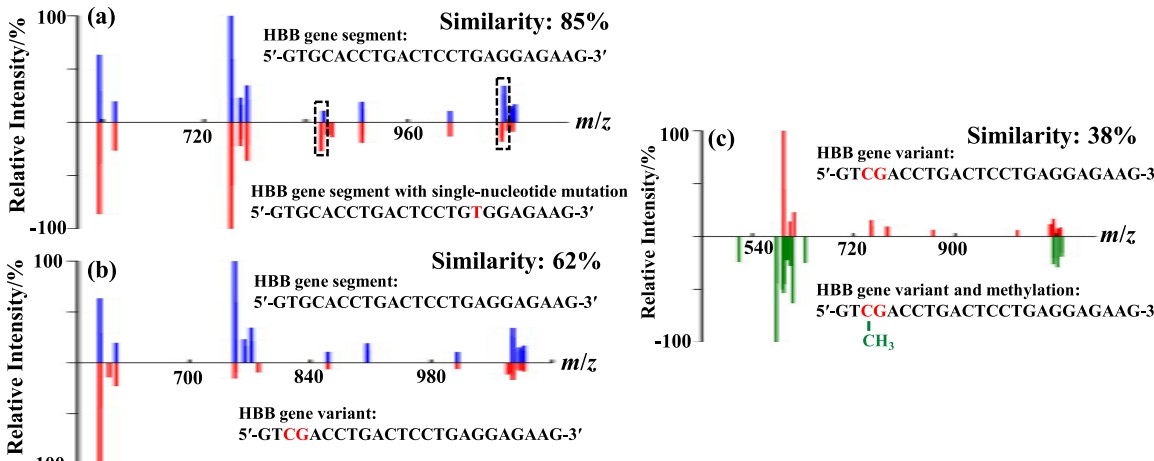

**Fig. 6.** Fragment mass pattern comparisons between: (*a*) the normal human hemoglobin beta chain (HBB) gene segment (5′-GTGCACCTGACTCCTGAGGAGAAG-3′) (blue) and the segment with a single-nucleotide mutation (5′-GTGCACCTGACTCCTGTGGAGAAG-3′) responsible for sickle cell anemia (red), (*b*) the normal HBB gene segment (blue) and the segment (5′-GTCGACCTGACTCCTGAGGAGAAG-3′) with a reversed order of two adjacent nucleotides (red), and (*c*) the gene segment (5′-GTCGACCTGACTCCTGAGGAGAAG-3′) with a reversed order of two adjacent nucleotides (red) and the segment (5′-GTCCH3GACCTGACTCCTGAGGAGAAG-3′) with a methylation of cytosine at CpG (green).

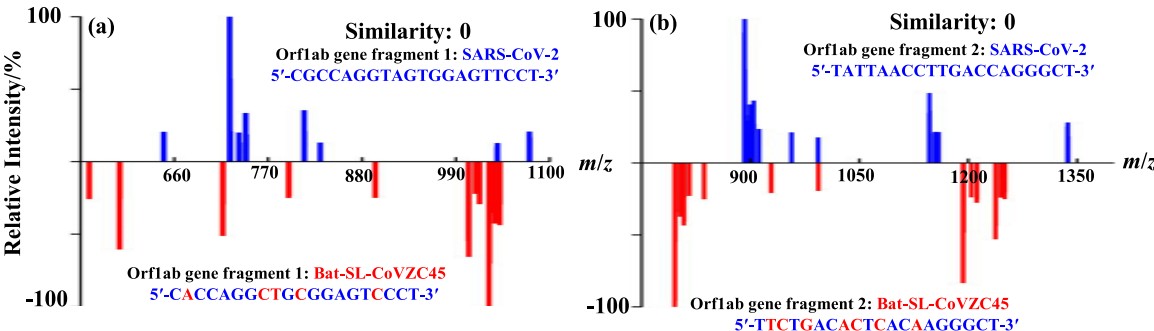

**Fig. 7.** Fragment mass pattern comparisons of the Orf1ab gene segments between: (*a*) segment 1 (5′-CGCCAGGTAGTGGAGTTCCT-3′) from SARS-CoV-2 (accession MN908947) (blue) and segment 1 (5′-CACCAGGCTGCGGAGTCCCT-3′) from SARS-like coronavirus of bat-SL-CoVZC45 (accession MG772933) (red), and (*b*) segment 2 (5′-TATTAACCTTGACCAGGGCT-3′) from SARS-CoV-2 (blue) and segment 2 (5′-TTCTGACACTCACAAGGGCT-3′) from Bat-SL-CoVZC45 (red).

discrimination capability of DMFP, several variants of HBB gene segment having the sequence of 5′-GTGCACCTGACTCCT-GAGGAGAAG-3′ were designed. In Fig. 6*a*, the sequence with a typical single-nucleotide mutation from the base of A to T (5′-GTGCACCTGACTCCTGTGGAGAAG-3′) responsible for sickle cell anemia (Guo *et al.*, 2010) provides a fragment mass pattern with a similarity of 85% compared to that of HBB gene segment. As marked by dashed square lines in Fig. 6*a*, two triple-charged peaks at *m/z* 860 and 1074 shifted to *m/z* 857 and 1071, respectively, corresponding to the mutation of base A to base T in the HBB gene segment. If we set the cutoff value of similarity at 90% as explained above, two DNA samples generating fragment mass patterns with a similarity larger than 90% were considered the same in sequence. Therefore, sickle cell anemia with a single-nucleotide mutation of the HBB gene segment could be distinguished from the normal sequence.

Though the standard gel assay could not provide sufficiently high resolution to resolve such a small change of molecular weight caused by single-nucleotide mutation, it could still be ascertained by direct mass detection with a high-resolution mass spectrometer. Motivated by this problem, we changed the order of two adjacent bases in HBB gene segment but with the same molecular weights (5′-GTCGACCTGACTCCTGAGGAGAAG-3′) and a similarity as

low as 62% was obtained with a significant change of the relative intensities of the fragments, as shown in Fig. 6*b*. This result showed the unique advantage of our approach over gel assay or direct mass detection. Similarly, the methylation modification of cytosine at CpG that is associated with a number of key life processes, such as aging and carcinogenesis (Robertson, 2005), could also be identified with a quite low similarity of 38% using our approach, as shown in Fig. 6*c*.

### *Discrimination of Orf1ab gene segments from SARS-CoV-2 (accession MN908947) and SARS-like coronavirus of bat-SL-CoVZC45 (accession MG772933)*

The identification of a coronavirus is typically done in three steps: (1) RNA extraction, (2) reverse transcription and (3) quantitative real-time PCR (qPCR) amplification of the resulting DNA. To show that our approach may have potential capability for practical applications, such as quick differentiation or identification of microorganisms, two short segments were extracted from the Orf1ab gene of SARS-CoV-2, which are normally selected as the unique targets for amplification and hybridisation in qPCR (Chu *et al.*, 2020), and compared to their corresponding gene segments of SARS-like coronavirus, bat-SL-CoVZC45 (Xiong *et al.*, 2020). Their

sequences are presented in Fig. 7. Though the homologous rate between SARS-CoV-2 and bat-SL-CoVZC45 is reported as high as 96% (Boni *et al.*, 2020), our approach of DMFP could provide totally different fragment mass patterns for the two selected segments with similarities both at 0%, respectively, as shown in Fig. 7*a*, *b*. These results suggest to us that DMFP may have wide applicability in simple, fast and economical microorganism genetic discrimination or identification.

## Reaction mechanism

It is natural to inquire about what factors cause the digestion of DNA by an endonuclease in aqueous microdroplets to be so remarkably accelerated compared to the rate of digestion in bulk solution. At the present time, no definitive simple answer exists. Reviews (Wei *et al.*, 2020) of past work pointed to several characteristics in water microdroplets that may aid reaction, and this topic is a matter of active continuing investigation. There is compelling evidence that reactions occur at the air-water interface, and at this boundary there exists a large electric field (Xiong *et al.*, 2020) whose origin is controversial. The suggestion has been made that this electric field arises from preferential adsorption of anionic surfactant impurities (Uematsu *et al.*, 2019), whereas other work suggests that the electric field at the water–air interface comes from the intrinsic polar nature of water (Doyle *et al.*, 2019). Whatever its origin, this electric field causes a strongly enhanced concentration of reactants at the interface (Xiong *et al.*, 2020), and creates a gradient of the concentration of ionic species throughout the microdroplet (Chamberlayne and Zare, 2020). It also causes reactants to become aligned (Zhou *et al.*, 2018), and internal rotation to be markedly reduced (Kang *et al.*, 2020). In addition, partial desolvation of reactants occurs at the interface, which can dramatically increase reactivity (Basuri *et al.*, 2020). These factors are interrelated and may account for why some reactions that are thermodynamically forbidden in bulk solution become allowed in water microdroplets(Nam *et al.*, 2017). Clearly, more research is needed to develop an understanding of which factors are most important in causing the remarkable acceleration of DNA digestion in aqueous microdroplets but there seems to be no denying that microdroplet chemistry is vastly different from the chemistry observed in bulk solution.

## Conclusions

Enzymatic digestion of proteins and therapeutic antibodies proven to be accelerated significantly from overnight to millisecond in microdroplets previously. In this study, we digest DNA with DNase I in microdroplets and found that the fragment mass pattern of a specific DNA sequence is stable and reproducible even under different experimental conditions. These results encouraged us to develop an approach called DNA fragment mass fingerprinting to discriminate different DNA sequences. Various examples were demonstrated with low similarities (<90%) between the fragment mass patterns of human hemoglobin beta chain gene segment and its several variants. Two Orf1ab gene segments of SARS-CoV-2 and bat-SL-CoVZC45 were also compared with similarities of 0 between their fragment mass patterns, showing the potential utility of our approach in simple and fast discrimination/identification of microorganisms. This work is an addition to recent interests in using mass spectrometry to study enzymatic reactions (Yan *et al.*, 2017; Hamilton *et al.*, 2020; Morato *et al.*, 2020; Zhong *et al.*, 2020; Zhao *et al.*, 2021).

**Acknowledgements.** This work has been supported by the Scientific Research Startup Foundation (IDH1615113) of Fudan University and the National Natural Science Foundation of China (21974027).

**Open Peer Review.** To view the open peer review materials for this article, please visit http://dx.doi.org/10.1017/qrd.2021.2.

**Supplementary Materials.** To view supplementary material for this article, please visit http://dx.doi.org/10.1017/qrd.2021.2.

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
