## [Reviewer Report]

*Comments to Author*: R. Graham Cooks and Nicolás M. Morato, Department of Chemistry, Purdue University, West Lafayette, IN 47907cooks@purdue.edunmoratog@purdue.edu

This study by Zhong, Hao and Zare is timely in view of the increased interest in using new methods of mass spectrometry for studies in enzymology that is evident in work over the past half dozen years.One instance is the work of Hamilton et al. [1] on functional mass spectrometry imaging of lipase activity by blotting tissue sections with phospholipidase. The MS image recorded from the blot provide a high-resolution map that reveals the location and identity of particular enzyme. Another example is the use of ambient ionization (ionization without sample preparation in the native environment) to directly measure aspartate aminotransferase activity from serum by direct product analysis. [2] These experiments used paper spray and were useful down to 60 µg/mL. Another example of the use novel instrumentation in enzyme MS is high throughput screening of substrate and product from the enzyme reaction mixture. [3] For example, determination for acetylcholinesterase without sample workup at 0.3 s per sample, to determine kinetic parameters, rapid inhibitor screening, and inhibition-reactivation assays.

Not only is the work of Xiaoqin Zhong, Hao Chen and Richard Zare timely, it is also dramatic in that it shows large increases in enzymatic reaction rates in droplets relative to bulk. This observation, made for proteases by some of the same authors (their refs 3 and 4), is demonstrated in this work to extend to nucleases. Dramatic increases in rates of simple organic reactions in droplets are well-established (their ref 1) and the underlying factors are beginning to become clearer with key experimental observations on the role of the interface (their ref 31) and of strong interfacial electric fields (their ref 28) associated with interfacial water.The authors address the question of why enzyme reactions are accelerated in the context of these same factors in their section titled reaction mechanism.

The paper demonstrates an exceptional ability (especially relative to conventional gel methods) to differentiate closely related sequences. However, the repeated claim of identification (starting with the title) is not supported if identification is taken to mean (as usual) de novo ID. Differential and identification are quite different capabilities but the authors often combine these terms here without giving evidence for identification.

The claim that discrimination if better than in 'conventional mass spectrometry' is valid but only if conventional mass spectrometry is taken to exclude the use of tandem mass spectrometry. The identification of oligonucleotide segments by tandem mass spectrometry has been known for many years [4] even though it does not have the speed and wide use enjoyed by protein MS/MS analysis (top down for full proteins, bottom up for bottom up methods used after enzymatic digestion).

The foregoing discussion occasions a comment about nomenclature. In mass spectrometry, a fragment (especially a fragment ion) refers to the product (ion) produced by dissociation of a precursor ion.Nucleic acid science also talks of fragments, referring to cleavage products generated enzymatically. In this study the terms fragment mass patterns (and variants of this) will be highly confusing to the MS community for whom the term has a very different meaning. If collisional or electron transfer / attachment methods begin to be used in conjunction with accelerated enzymatic reactions the confusion will be nearly perfect.Is it too much to ask the authors not to adopt their chosen nomenclature in the interests of clarity of future communication?

The microsecond time scale of the enzymatic reaction is demonstrated but the wording is may sometimes be too strong, as in "ultrafast digestion … could be completed within microseconds" (p. 3). The digestion yes, but no other part of the experiment.

Explanation of the numerical part of the statement "very high homologous rate of 96%...." would help some readers.

The comparison with the bulk experiment (p. 3) allows a rate acceleration factor, used in comparisons of rate constants in microdroplets and in bulk for organic reactions of small molecules (their ref. 1), to be given explicitly here too. These numbers might be useful as the mechanistic question is subjected to increasing scrutiny.

The effect of travel distance on the extent of reaction is a good method of following kinetics provided the droplet does not evaporate significantly. This is likely the case for aqueous droplets. The data (Fig. 2) show a significant effect of distance on relative intensity but what is missing is any information on absolute signals as a function of distance. This information needs to be provided: if the fall off with distance is very large it might include differential transmission effects which are not evident but which will affect the interpretation.

It might be useful to also express the enzyme concentration in molar units to allow better comparison with the substrate concentrations. The authors indicate that for a typical experiment they use 10 µM DNA substrate and 10 µg/mL DNase I, (MW. ca. 35,000 Da), giving concentrations in the ratio of 35:1. This is a high ratio of enzyme to substrate and deserves discussion. Especially considering that a 1:10 reduction in enzyme concentration gave no products at all. Note that no data at this enzyme concentration is shown for the bulk reaction (either 1 h or overnight) for comparison.Clarification would be useful on the description of the enzyme utilized: it is said to be >500 kU/mg: does this mean 500,000 U/mg or 500 Kunitz Units/mg?

The comparison of the microdroplet reaction vs. bulk after 1 h and overnight seems to be incorrectly interpreted as suggesting that the "fragment mass pattern obtained by digestion in microdroplets keeps stable". It is expected that as the bulk reaction progresses overnight it would give more lower mass fragments compared to 1 h (which would favor larger mass fragments). This is what the authors observed but not what they state. These results could be used to conclude that the microdroplet accelerated reaction proceeds as much as the 1 h bulk reaction but less than the overnight bulk reaction, under the conditions tested.

During the comparison of the mass patterns obtained for the two SARS-CoV-2 and bat-SL-CoVZC45 segments a similarity score of 0% is presented in both of the two cases shown. This seems surprising, as it would mean there is no match at any single m/z value. Is that the case? Is this just a rounding error? If so, perhaps it is worth keeping some decimal places. Another important question concerns the mass ranges used for the calculation of the scores. At least visually in the ranges shown (Fig. 7) one can observe one peak matching in each case. Evidently, and again at least in the ranges shown, the similarity scores will be quite low, but not exactly zero.

The comparison to PCR stated in the introduction seems incorrect. Quantitative (real-time) PCR is extensively used in genotyping, even in multiplexed platforms, and it can be used to detect down to single nucleotide polymorphisms. The microdroplet method presented, despite being impressive in terms of speed, is demonstrated for a fairly high concentration (10 µM) of pure DNA segments. PCR, by design, deals with traces of genetic material thanks to the inherent amplification of the technique, and of course uses much more complex samples, as well as longer nucleic acid sequences.

[1] Brett R. Hamilton, David L. Marshall, Nicholas R. Casewell, Robert A. Harrison, Stephen J. Blanksby and Eivind A. B. Undheim "Mapping Enzyme Activity on Tissue by Functional Mass Spectrometry Imaging" Angew. Chem. 59 (2020) 3855-3858 https://doi.org/10.1002/anie.201911390

[2] Xin Yan, Xin Li, Chengsen Zhang, Yang Xu and R. Graham Cooks "Ambient Ionization Mass Spectrometry Measurement of Aminotransferase Activity" J. Amer. Soc. Mass Spectrom. 28 (2017) 1175-1181https://doi.org/10.1021/jasms.8b05557

[3] Nicolás M. Morato, Dylan T. Holden and R. Graham Cooks, "High‐Throughput Label‐Free Enzymatic Assays Using Desorption Electrospray‐Ionization Mass Spectrometry" Angew. Chem. Int. Ed. 59 (2020) 20459-20464https://doi.org/10.1002/anie.202009598

[4] Jin Wu and Scott A. McLuckey "Gas-phase fragmentation of oligonucleotide ions" Int. J. Mass Spectrom.237 (2004) 197-241 https://doi.org/10.1016/j.ijms.2004.06.014

---

## [Reviewer Report]

*Comments to Author*: The reaction acceleration in mircrodroplets has been widely recognized (mainly by the authors of this paper) in small molecule reactions and nanomaterial synthesis. The phenomenon is intriguing both in chemistry and physics, and might find its applications in biology. Here we are now: this work is expanding the field to the more complicated large biomolecules (accelerated digestion of DNA), adding to the diversity of the library of microdroplet chemistry.

I urge the publication of this work without any revisions.

---

## [Reviewer Report]

*Comments to Author*: This work by Zhong et. al. reports an ultrafast method to enzymatically digest DNA in aqueous microdroplets, so different DNA sequences can be distinguished by comparing the similarity of the fragments generated from the DNA sequences. The authors call this method DNA fragment mass fingerprinting (DMFP). The method has been demonstrated in real samples including the sickle cell anemia mutation and Orf1ab gene of two similar coronaviruses. The authors have published many remarkable works in microdroplet acceleration including but not limited to organic reactions, nanoparticle formation, redox reactions, and enzymatic peptide digestions. This work fills the gap of accelerated DNA digestion using microdroplets. The DMFP method is much more sensitive than gel electrophoresis in detecting tiny sequence differences such as single-nucleotide mutation, and is much faster than the traditional genotyping method with no requirement of expensive reagents. This is another high-quality work from Zare and Hao groups and I am in strong favor of supporting the work for publication in QRB Discovery. I suggest the authors address the minor issues listed below before eventual publication.

1.The authors studied base-preferential cleavage by DNase I and concluded that DNase I shows the highest preference for cleaving at the T base, while A and C are strongly resistant to fragmentation by DNase I. This was completely different from what we knew about DNase I that does not show specificity to base or sequence when cleaving DNA. The conclusion was based on the digestion extents of a series of different DNA sequences in Table S1. However, no identification of fragments was provided and no specific positions of cleavage were identified. I'm not quire convinced that DNase I "has base-preferential cleavage" at those base positions.

2.The authors did not identify/label fragments between m/z 500-900 for the DNA sequence 5′-TAGTCAGCTACGGCTAGA-3′ after digestion. Also, no assignment of the fragments was provided in Figure 6 or 7.

3.In Figure 2, the authors indicated the cleavage at TC.There were a few MS peaks between fragment-1 and fragment-2, and their intensities were not low. Can the authors comment on what are those peaks?

4.In Figure 5, c shows the similarity between 1h bulk reaction and microdroplet reaction was 99%. What about the bulk reaction of less than 1 h? The authors use a cutoff value of 90% for similarity evaluation. When does the similarity reach the cut-off value between the microdroplet reaction and the bulk reaction?

5.It seems that the method cannot fragment DNA sequence completely and the fragmentation highly depends on the cleavage sites in a DNA sequence. Is there any situation that two different DNA sequences cannot be differentiated by DMFP because the differences are not shown at the cleavage sites? Will this lead to false-negative results?

6.Can the authors use the same m/z scale for Figure 3c,d? so that similarity comparison can be clearer.

7.Both the full name "DNA fragment mass fingerprinting" and the abbreviation "DMFP" appear many times in the main text. Please use the abbreviation after the initial definition.

8.In the sentence "Therefore, a longer travel distance of microdroplets result in…", "result" should be "results".

9.In the sentence "The digestion time in the microdroplets…on the previous reported microdroplet velocity of 80 meter per second…", "80 meter per second" should be "80 meters per second".

---

## [Reviewer Report]

*Comments to Author*: Reviewer #1: The reaction acceleration in mircrodroplets has been widely recognized (mainly by the authors of this paper) in small molecule reactions and nanomaterial synthesis. The phenomenon is intriguing both in chemistry and physics, and might find its applications in biology. Here we are now: this work is expanding the field to the more complicated large biomolecules (accelerated digestion of DNA), adding to the diversity of the library of microdroplet chemistry.

I urge the publication of this work without any revisions.

Reviewer #2: R. Graham Cooks and Nicolás M. Morato, Department of Chemistry, Purdue University, West Lafayette, IN 47907cooks@purdue.edunmoratog@purdue.edu

This study by Zhong, Hao and Zare is timely in view of the increased interest in using new methods of mass spectrometry for studies in enzymology that is evident in work over the past half dozen years.One instance is the work of Hamilton et al. [1] on functional mass spectrometry imaging of lipase activity by blotting tissue sections with phospholipidase. The MS image recorded from the blot provide a high-resolution map that reveals the location and identity of particular enzyme. Another example is the use of ambient ionization (ionization without sample preparation in the native environment) to directly measure aspartate aminotransferase activity from serum by direct product analysis. [2] These experiments used paper spray and were useful down to 60 µg/mL. Another example of the use novel instrumentation in enzyme MS is high throughput screening of substrate and product from the enzyme reaction mixture. [3] For example, determination for acetylcholinesterase without sample workup at 0.3 s per sample, to determine kinetic parameters, rapid inhibitor screening, and inhibition-reactivation assays.

Not only is the work of Xiaoqin Zhong, Hao Chen and Richard Zare timely, it is also dramatic in that it shows large increases in enzymatic reaction rates in droplets relative to bulk. This observation, made for proteases by some of the same authors (their refs 3 and 4), is demonstrated in this work to extend to nucleases. Dramatic increases in rates of simple organic reactions in droplets are well-established (their ref 1) and the underlying factors are beginning to become clearer with key experimental observations on the role of the interface (their ref 31) and of strong interfacial electric fields (their ref 28) associated with interfacial water.The authors address the question of why enzyme reactions are accelerated in the context of these same factors in their section titled reaction mechanism.

The paper demonstrates an exceptional ability (especially relative to conventional gel methods) to differentiate closely related sequences. However, the repeated claim of identification (starting with the title) is not supported if identification is taken to mean (as usual) de novo ID. Differential and identification are quite different capabilities but the authors often combine these terms here without giving evidence for identification.

The claim that discrimination if better than in 'conventional mass spectrometry' is valid but only if conventional mass spectrometry is taken to exclude the use of tandem mass spectrometry. The identification of oligonucleotide segments by tandem mass spectrometry has been known for many years [4] even though it does not have the speed and wide use enjoyed by protein MS/MS analysis (top down for full proteins, bottom up for bottom up methods used after enzymatic digestion).

The foregoing discussion occasions a comment about nomenclature. In mass spectrometry, a fragment (especially a fragment ion) refers to the product (ion) produced by dissociation of a precursor ion.Nucleic acid science also talks of fragments, referring to cleavage products generated enzymatically. In this study the terms fragment mass patterns (and variants of this) will be highly confusing to the MS community for whom the term has a very different meaning. If collisional or electron transfer / attachment methods begin to be used in conjunction with accelerated enzymatic reactions the confusion will be nearly perfect.Is it too much to ask the authors not to adopt their chosen nomenclature in the interests of clarity of future communication?

The microsecond time scale of the enzymatic reaction is demonstrated but the wording is may sometimes be too strong, as in "ultrafast digestion … could be completed within microseconds" (p. 3). The digestion yes, but no other part of the experiment.

Explanation of the numerical part of the statement "very high homologous rate of 96%...." would help some readers.

The comparison with the bulk experiment (p. 3) allows a rate acceleration factor, used in comparisons of rate constants in microdroplets and in bulk for organic reactions of small molecules (their ref. 1), to be given explicitly here too. These numbers might be useful as the mechanistic question is subjected to increasing scrutiny.

The effect of travel distance on the extent of reaction is a good method of following kinetics provided the droplet does not evaporate significantly. This is likely the case for aqueous droplets. The data (Fig. 2) show a significant effect of distance on relative intensity but what is missing is any information on absolute signals as a function of distance. This information needs to be provided: if the fall off with distance is very large it might include differential transmission effects which are not evident but which will affect the interpretation.

It might be useful to also express the enzyme concentration in molar units to allow better comparison with the substrate concentrations. The authors indicate that for a typical experiment they use 10 µM DNA substrate and 10 µg/mL DNase I, (MW. ca. 35,000 Da), giving concentrations in the ratio of 35:1. This is a high ratio of enzyme to substrate and deserves discussion. Especially considering that a 1:10 reduction in enzyme concentration gave no products at all. Note that no data at this enzyme concentration is shown for the bulk reaction (either 1 h or overnight) for comparison.Clarification would be useful on the description of the enzyme utilized: it is said to be >500 kU/mg: does this mean 500,000 U/mg or 500 Kunitz Units/mg?

The comparison of the microdroplet reaction vs. bulk after 1 h and overnight seems to be incorrectly interpreted as suggesting that the "fragment mass pattern obtained by digestion in microdroplets keeps stable". It is expected that as the bulk reaction progresses overnight it would give more lower mass fragments compared to 1 h (which would favor larger mass fragments). This is what the authors observed but not what they state. These results could be used to conclude that the microdroplet accelerated reaction proceeds as much as the 1 h bulk reaction but less than the overnight bulk reaction, under the conditions tested.

During the comparison of the mass patterns obtained for the two SARS-CoV-2 and bat-SL-CoVZC45 segments a similarity score of 0% is presented in both of the two cases shown. This seems surprising, as it would mean there is no match at any single m/z value. Is that the case? Is this just a rounding error? If so, perhaps it is worth keeping some decimal places. Another important question concerns the mass ranges used for the calculation of the scores. At least visually in the ranges shown (Fig. 7) one can observe one peak matching in each case. Evidently, and again at least in the ranges shown, the similarity scores will be quite low, but not exactly zero.

The comparison to PCR stated in the introduction seems incorrect. Quantitative (real-time) PCR is extensively used in genotyping, even in multiplexed platforms, and it can be used to detect down to single nucleotide polymorphisms. The microdroplet method presented, despite being impressive in terms of speed, is demonstrated for a fairly high concentration (10 µM) of pure DNA segments. PCR, by design, deals with traces of genetic material thanks to the inherent amplification of the technique, and of course uses much more complex samples, as well as longer nucleic acid sequences.

[1] Brett R. Hamilton, David L. Marshall, Nicholas R. Casewell, Robert A. Harrison, Stephen J. Blanksby and Eivind A. B. Undheim "Mapping Enzyme Activity on Tissue by Functional Mass Spectrometry Imaging" Angew. Chem. 59 (2020) 3855-3858 https://doi.org/10.1002/anie.201911390

[2] Xin Yan, Xin Li, Chengsen Zhang, Yang Xu and R. Graham Cooks "Ambient Ionization Mass Spectrometry Measurement of Aminotransferase Activity" J. Amer. Soc. Mass Spectrom. 28 (2017) 1175-1181https://doi.org/10.1021/jasms.8b05557

[3] Nicolás M. Morato, Dylan T. Holden and R. Graham Cooks, "High‐Throughput Label‐Free Enzymatic Assays Using Desorption Electrospray‐Ionization Mass Spectrometry" Angew. Chem. Int. Ed. 59 (2020) 20459-20464https://doi.org/10.1002/anie.202009598

[4] Jin Wu and Scott A. McLuckey "Gas-phase fragmentation of oligonucleotide ions" Int. J. Mass Spectrom.237 (2004) 197-241 https://doi.org/10.1016/j.ijms.2004.06.014

Reviewer #3: This work by Zhong et. al. reports an ultrafast method to enzymatically digest DNA in aqueous microdroplets, so different DNA sequences can be distinguished by comparing the similarity of the fragments generated from the DNA sequences. The authors call this method DNA fragment mass fingerprinting (DMFP). The method has been demonstrated in real samples including the sickle cell anemia mutation and Orf1ab gene of two similar coronaviruses. The authors have published many remarkable works in microdroplet acceleration including but not limited to organic reactions, nanoparticle formation, redox reactions, and enzymatic peptide digestions. This work fills the gap of accelerated DNA digestion using microdroplets. The DMFP method is much more sensitive than gel electrophoresis in detecting tiny sequence differences such as single-nucleotide mutation, and is much faster than the traditional genotyping method with no requirement of expensive reagents. This is another high-quality work from Zare and Hao groups and I am in strong favor of supporting the work for publication in QRB Discovery. I suggest the authors address the minor issues listed below before eventual publication.

1.The authors studied base-preferential cleavage by DNase I and concluded that DNase I shows the highest preference for cleaving at the T base, while A and C are strongly resistant to fragmentation by DNase I. This was completely different from what we knew about DNase I that does not show specificity to base or sequence when cleaving DNA. The conclusion was based on the digestion extents of a series of different DNA sequences in Table S1. However, no identification of fragments was provided and no specific positions of cleavage were identified. I'm not quire convinced that DNase I "has base-preferential cleavage" at those base positions.

2.The authors did not identify/label fragments between m/z 500-900 for the DNA sequence 5′-TAGTCAGCTACGGCTAGA-3′ after digestion. Also, no assignment of the fragments was provided in Figure 6 or 7.

3.In Figure 2, the authors indicated the cleavage at TC.There were a few MS peaks between fragment-1 and fragment-2, and their intensities were not low. Can the authors comment on what are those peaks?

4.In Figure 5, c shows the similarity between 1h bulk reaction and microdroplet reaction was 99%. What about the bulk reaction of less than 1 h? The authors use a cutoff value of 90% for similarity evaluation. When does the similarity reach the cut-off value between the microdroplet reaction and the bulk reaction?

5.It seems that the method cannot fragment DNA sequence completely and the fragmentation highly depends on the cleavage sites in a DNA sequence. Is there any situation that two different DNA sequences cannot be differentiated by DMFP because the differences are not shown at the cleavage sites? Will this lead to false-negative results?

6.Can the authors use the same m/z scale for Figure 3c,d? so that similarity comparison can be clearer.

7.Both the full name "DNA fragment mass fingerprinting" and the abbreviation "DMFP" appear many times in the main text. Please use the abbreviation after the initial definition.

8.In the sentence "Therefore, a longer travel distance of microdroplets result in…", "result" should be "results".

9.In the sentence "The digestion time in the microdroplets…on the previous reported microdroplet velocity of 80 meter per second…", "80 meter per second" should be "80 meters per second".